# Macmoondongtang modulates Th1-/Th2-related cytokines and alleviates asthma in a murine model

**Soon-Young Lee[1]☉, Bossng Kang[2]☉, So-Hyeon Bok[1], Seung Sik Cho[3]\*, Dae-Hun Park [1]\***

**1** College of Oriental Medicine, Dongshin University, Naju, Jeonnam Korea, **2** Department of Emergency Medicine, College of Medicine, Hanyang University, Guri, Gyunggi, Korea, **3** Department of Pharmacy, College of Pharmacy, Mokpo National University, Muan, Jeonnam Korea

☉ These authors contributed equally to this work.
\* sscho@mokpo.ac.kr (SSC); dhj1221@hanmail.net (DHP)

## Abstract

### Objective

*Macmoondongtang* has been used as a traditional medicine to treat pulmonary disease in Korea. However, the mechanism underlying its therapeutic effect has yet to be reported. In the present study, the role of *macmoondongtang* as a respiratory medicine, especially as an anti-asthmatic agent, has been attributed to the down-regulation of interleukin (IL)-4 and tumor necrosis factor (TNF)-α.

### Materials & methods

BALB/c mice were divided into five groups: control, asthma-induced control, dexamethasone treatment, treatment with 150 mg/kg *macmoondongtang*, and treatment with 1500 mg/kg *macmoondongtang*. To evaluate the anti-asthmatic effect of macmoondongtang, we investigated its suppressive or inhibitory effects against typical asthmatic changes such as differential cell count in bronchioalveolar fluid (BALF), serum IgE levels, lung morphology, expression of Th1/Th2 cell transcription factors such as T-bet and GATA-3, and Th1-/Th2-/Th17-related cytokines such as interferon (IFN)-γ, IL-12p40, IL-4, -5, -13, TNF-α, and IL-6. The active ingredients in macmoondongtang were further analyzed.

### Results

*Macmoondongtang* treatment down-regulated serum IgE level, a very important marker of hyper-responsiveness. It reversed typical morphological changes such as mucous hypersecretion, lung epithelial cell hyperplasia, and inflammatory cell infiltration near bronchioalveolar space and veins. *Macmoondongtang* significantly decreased neutrophil count in BALF, as well as reduced T-bet, IFN-γ, and TNF-α expression in the lung. It also showed a dose-dependent control of inflammatory cells in BALF, controlled the expression of *IL-12*, *IL-4*, and *IL-5* genes in the lung, and the protein expression of IL12p40, GATA-3, IL-4, IL-5, and IL-13. The component analysis revealed glycyrrhizin and liquiritin as the active ingredients.

**Data Availability Statement:** All relevant data are within the paper and its Supporting Information files.

**Funding:** This work was supported by the National Research Foundation of Korea (NRF) grant funded

by the Korean government (MSIP: Ministry of Science, ICT & Future Planning) (Grant Nos. NRF-2015R1D1A1A01059523 to D-HP and NRF-2017R1C1B5015187 to SSC). The funders had no role in study design, data collection and analysis, decision to publish, or preparation of the manuscript.

## Conclusions

*Macmoondongtang* treatment alleviates asthma symptoms and modulate the Th1-/Th2-related cytokines. Glycyrrhizin and liquiritin could be the major the active therapeutic components.

## Introduction

In 2013, the World Health Organization reported that at least 200 million individuals worldwide were diagnosed with asthma. The allergens were classified into two categories: indoor agents (such as pet dander, dust mites, tobacco smoke, and so forth) and outdoor agents (e.g., pollen, environmental pollutants, cold temperature, and so forth) [1]. Asthma is a type I allergy, and anaphylaxis. It is hard to completely eradicate and its symptoms are very diverse, ranging from cough to apnea. Patients with severe allergy may die from pulmonary obstruction and lack of gas exchange due to hypersecretion of mucus, overgrowth of pulmonary subepithelial cells, enlarged basement membrane, and accumulation of inflammatory cells near bronchioles and vessels [2,3].

Among several theories of asthma occurrence, the imbalance between Th1 and Th2 cells plays a significant role in increasing the risk [2]. Cytokines such as interferon gamma (IFN-γ) that induce the release of Th1 cells are associated with the severity of asthma [4]. Interleukin 12 (IL-12) is an important cytokine that mediates Th1 cell differentiation and decreases Th2 cell proliferation [5–7]. Th2 cells release specific cytokines such as IL-4, IL-5, IL-13, and tumor necrosis factor-alpha (TNF-α) while IL-6 and IL-1β are Th17-related cytokines [8]. IL-4 is induced by GATA-3, a transcription factor with several bioactive functions, including stimulation of Th2-related cytokine expression [9], regulation of IgE levels [10], and so on. The level of IL-6 in asthma patients is substantially higher than in normal persons [11]. As one of the important factors that regulate asthma severity, IL-6 induces IL-4 up-regulation and stimulates Th17 cell differentiation [12]. IL-13 is linked to morphological changes in respiratory organs and B cell activation [13–16]. TNF-α is synthesized by macrophages [17]. It recruits neutrophils and eosinophils [18], stimulates T cell activation [19], and finally establishes airway hyperresponsiveness [20].

Inhaled corticosteroids are usually used to inhibit clinical asthma symptoms such as cough, respiratory depression, etc [21]. Asthma occurrence is higher in early childhood and elderly population than in the young and grown-up groups [1]. Inhaled corticosteroids are associated with serious adverse effects such as growth inhibition [22], cataracts, glaucoma, hypertension, hyperlipidemia, peptic ulcers, myopathy, and immunological suppression [23]. Therefore, efforts to discover new anti-asthmatic agents are ongoing and trials to investigate the mode of action on suppression or inhibition against asthma occurrence among the traditional medicines-related pulmonary diseases have increased.

Several hundred years ago in Korea, a medical encyclopedia known as *Donguibogam* was composed [24]. *Donguibogam* introduced *macmoondongtang*, a traditional prescription for the treatment of respiratory illness. *Macmoondongtang* is composed of plants such as *Glycyrrhizae radix*, *Oryza sativa*, *Ziziphus jujube* Inermis, *Ophiopogon japonicas*, *Pinellia ternate*, and *Panax ginseng*. Although *macmoondongtang* has been used to treat pulmonary disease recently, its mode of action has yet to be reported. Therefore, the objective of this study was to analyze the possible anti-asthmatic effect of *macmoondongtang* and the underlying mechanism of action.

## Materials and methods

### Preparation of *macmoondongtang*

*Macmoondongtang* (Batch No. 14010, hot water extract-powder form) was manufactured by Hankuk Inspharm, Ltd. (Jeonnam, Korea) based on the prescription indicated in *Donguibogam* [24] and used in the present study. *Macmoondongtang* is generally available as an over-the-counter (OTC) drug. It contains *G. radix*, *O. sativa*, *Z. jujube* Inermis, *O. japonicas*, *P. ternate*, and *P. ginseng*. The dosage used in this study was calculated based the drug composition. The human equivalent dose (HED) for mice is 12.3 and the safety factor is 10×[25]. In this study, we applied 1/100 and 1/10 dosages for mouse based on therapeutic human dosage.

### Identification of anti-asthmatic compounds in *macmoondongtang*

To analyze the levels of anti-asthmatic compounds in *macmoondongtang*, appropriate amounts of glycyrrhizin and liquiritin were accurately weighed and dissolved in methanol in 100 mL volumetric flasks to obtain stock solutions of 100 μg/mL. Solutions were subsequently serially diluted two-fold to 3.125 μg/mL. A sample measuring 0.5 g was dissolved in 10 mL methanol and sonicated to expedite the dissolution of particles. Subsequently, 1 mL was transferred to a volumetric flask and diluted with 9 mL of mobile phase A. The final concentration of *macmoondongtang* was 50 mg/mL. HPLC analysis was performed using an Alliance 2695 HPLC system (Waters, Milford, MA, USA) equipped with a photodiode array detector. A reverse phase column (C18, 5 μm, 150 mm × 5 mm) was used with a mobile phase consisting of a mixture of solvent A (acetonitrile) and B (0.2% acetic acid). The mobile phase conditions for glycyrrhizin were as follows: 0–35 min, gradient elution from 15/85 to 35/65 v/v; 35–50 min, gradient elution from 35/65 to 100/0 v/v; and 50–55 min, from 100/0 to 15/85 v/v. The mobile phase conditions for liquiritin were as follows: 0–8 min, gradient elution from 10/90 to 20/80 v/v; 8–20 min, gradient elution from 20/80 to 25/75 v/v; 20–21 min, from 25/75 to 100/ v/v; and 21–30 min, from 100/0 to 10/90 v/v. The flow rate was 1.0 mL/min, and the injection volume was 10 μL.

### Animal experiments

The animal study investigating the anti-asthmatic effect of *macmoondongtang* was followed by a confirmative study. For each study, 35 female BALB/c mice were purchased from Samtako Korea (Osan, Korea) and divided into five groups according to the treatment: vehicle control (tap water), ovalbumin (OVA)-treated group, OVA-treated group exposed to 1 mg/kg dexamethasone, OVA-treated group exposed to 150 and 1500 mg/kg *macmoondongtang*. The following doses were injected intraperitoneally into all animals except the control group: 20 μg OVA (Sigma-Aldrich, St. Louis, MO, USA) and 1 mg aluminum hydroxide hydrate (Sigma-Aldrich) in 500 μL saline. A week after injection of the second booster dose, all animals except the control group were exposed to OVA including a first inhalation dose for 5 days after OVA challenge. The animals were treated orally with tap water, dexamethasone, and 2 doses of *macmoondongtang*.

### Ethics statement

This animal study was conducted following approval by the Institutional Animal Care and Use Committee of Dongshin University (Animal Study Approval No. 2014-08-02).

## Bronchoalveolar fluid (BALF) and serum analysis

Bronchioalveolar fluid (BALF) was collected from all mice under anesthesia on a day after the final treatment. BALF was collected using 0.4 mL cold phosphate-buffered saline (PBS). The white blood cell (WBC) count and differential blood count were performed with the collected fluid using Hemavet Multispecies Hematology System (Drew Scientific Inc, Waterbury, CT, USA). Inflammatory cells in BALF were stained with Diff-Quick method [26].

Serum IgE levels were evaluated using ELISA.

## Histopathological analysis

In order to observe the morphological changes, lungs were fixed in formaldehyde, dehydrated in ethanol, embedded in paraffin, sectioned, and stained with hematoxylin and eosin. In order to compare the levels of released glycoprotein, periodic acid Schiff (PAS) stain was used. Subsequently, the images were acquired with Axioscope A1 (Carl Zeiss, Gottingen, Germany).

## Reverse transcription polymerase chain reaction (RT-PCR) and enzyme-linked immunoassay (ELISA)

In order to evaluate changes in the cDNA levels of IFN-$\gamma$, IL-12p40, IL-4, IL-5, IL-13, TNF-$\alpha$, and IL-6, which are related to asthma induction, RT-PCR analysis was conducted as reported in our previous study protocol (Lee et al., 2017). Total RNA was extracted from the lung using the RNeasy Mini Kit (Qiagen, Hilden, Germany) according to the manufacturer's instructions. Total RNA (100 ng) was used as a template for the reaction. Primers were synthesized for RT-PCR as shown in Table 1. The RT-PCR cycles consisted of denaturation at 95˚C for 5 s and annealing/extension at 65˚C for 30 s for 40 cycles using with QTOWER2.2. (Analytik Jena AG, Thuringia, Germany). The serum levels of IgE were measured using a specific mouse IgE enzyme-linked immunosorbent assay kit (BD bioscience, 555248, San Jose, CA, USA) according to the manufacturer's protocols.

**Table 1. Primer sequences for RT-PCR.**

| Genes | Primer sequences | |
|---|---|---|
| IFN-$\gamma$ | Forward | 5′–GGCCATCAGCAACAACATAAG–3′ |
| | Reverse | 5′–GTTGACCTCAAACTTGGCAATAC–3′ |
| IL-12p | Forward | 5′–GGACCAAAGGGACTATGAGAAG–3′ |
| | Reverse | 5′–CTTCCAACGCCAGTTCAATG–3′ |
| IL-4 | Forward | 5′–ACAGGAGAAGGGACGCCAT–3′ |
| | Reverse | 5′–GAAGCCCTACAGACGAGCTCA–3′ |
| IL-5 | Forward | 5′–TGCATCAGGGTCTCAAGTATTC–3′ |
| | Reverse | 5′–GGATGCTAAGGTTGGGTATGT–3′ |
| IL-13 | Forward | 5′–CAGCCCTCAGCCATGAAATA–3′ |
| | Reverse | 5′–CTTGAGTGTGTAACAGGCCATTCT–3′ |
| IL-6 | Forward | 5′–GATAAGCTGGAGTCACAGAAGG–3′ |
| | Reverse | 5′–TTGCCGAGTAGATCTCAAAGTG–3′ |
| TNF-$\alpha$ | Forward | 5′–CTGAGTTCTGCAAAGGGGAGAG–3′ |
| | Reverse | 5′–CCTCAGGGAAGAATCTGGAAAG–3′ |
| GAPDH | Forward | 5′–GTGGAGTCATACTGAACATGTAG–3′ |
| | Reverse | 5′–AATGGTGAAGGTCGGTGTG–3′ |

## Immunofluorescent analysis

In order to localize the expression of Th1 cell transcription factor, T-bet, and Th2 cell transcription factor, GATA-3 immunofluorescent analysis was conducted. Only four groups were evaluated: control, OVA, dexamethasone plus OVA treatment, and 1500 mg/kg *macmoondongtang* plus OVA treatment. Prior to the antibody binding step, the same materials were used for immunohistochemical analysis; however, rabbit anti-mouse T-bet (Biorbyt, orb7075, Cambridge, UK) or goat anti-mouse GATA-3 (OriGene, TA305795, Rockville, MD, USA) were used as primary antibodies for 1 h at room temperature. The slides were incubated for 2 h with FITC-conjugated anti-rabbit IgG (Jackson Immunoresearch, 315-095-003, West Grove, PA, USA) or Alexa Fluor® 555-conjugated anti-goat IgG (ThermoFisher Scientific, A-21127, Waltham, MA, USA). The cells were counterstained with DAPI (ThermoFisher Scientific, 62249, Waltham, MA, USA). The images were obtained using a K1-Fluo confocal microscope (Nanoscope System, Daejeon, Korea).

## Immunohistochemical analysis

In order to analyze the changes in the levels of asthma-related protein, immunohistochemistry was performed using antibodies such as IFN-γ (Santa Cruz, sc-74104), IL-12p40 (Santa Cruz, sc-57258), IL-4 (Santa Cruz, sc-73318), IL-5 (Santa Cruz, sc-7887), IL-13 (Santa Cruz, sc-1776), IL-6 (Santa Cruz, sc-1265), and TNF-α (BioVision, 3053R-100, Milpitas, CA, USA) and the dilution rate ranged from 1:100 to 1:200. The slides were incubated with a biotinylated pan-specific secondary antibody for 10 min and reacted with the streptavidin-peroxidase complex for 5 min (Vector Laboratories Universal Quick Kit, Burlingame, Canada). Signals were detected using 3,3-diaminobenzidine tetrahydrochloride substrate chromogen solution. Cells were counterstained with Mayer's hematoxylin. Subsequently, the cells were imaged using Axioscope A1 (Carl Zeiss).

## Statistical analysis

Results are expressed as the mean ± standard deviation (SD). Group differences were evaluated via one-way analysis of variance followed by Dunnett's multiple comparison tests. Significance was considered at $p < 0.01$ or $p < 0.05$.

# Results

### *Macmoondongtang* contains two anti-asthmatic markers: Glycyrrhizin and liquiritin

HPLC analyses were performed to identify the anti-asthmatic compounds and immune modulators in *macmoondongtang*. Typical HPLC chromatograms and their retention times are shown in Fig 1. Glycyrrhizin (0.38 ± 0.002%), and liquiritin (0.02 ± 0.0002%) were identified using HPLC analysis.

### *Macmoondongtang* suppresses ovalbumin-induced hyperproliferation of inflammatory cells and immune cells

OVA treatment significantly boosts the levels of inflammatory and immune cells such as white blood cells, eosinophils, and neutrophils in the lung [27]. *Macmoondongtang* induced a dose-dependent suppression of inflammatory cells, which were increased by OVA treatment (Fig 2A and S1 File). We used dexamethasone as a positive control during the treatment with macmoondongtang as it is generally used to manage acute asthma exacerbation [28]. As shown in

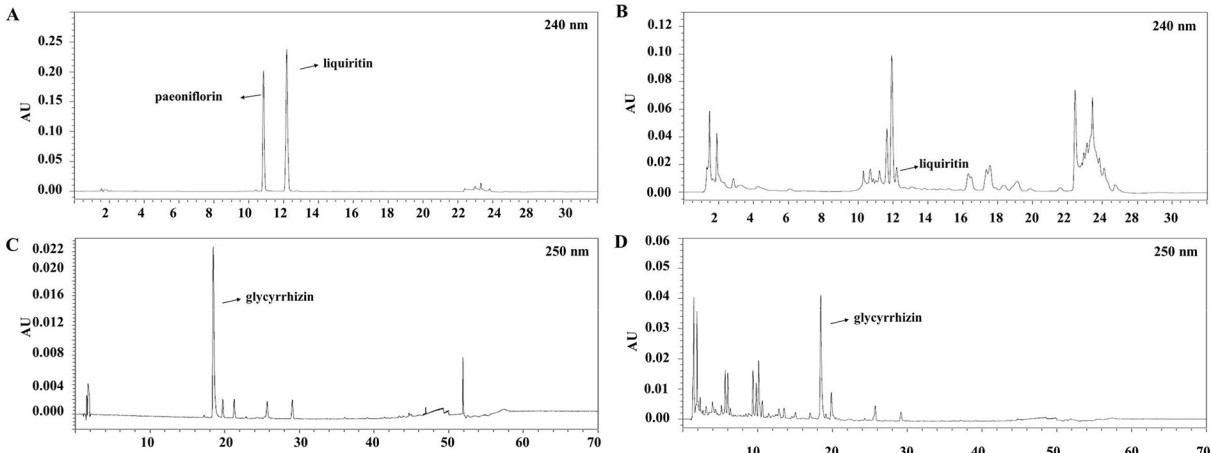

**Fig 1. HPLC results of *macmoondongtang*.** (A) standard- liquiritin and paeoniflorin; (B) *macmoondongtang*, 50 mg/mL, (C) standard-glycyrrhizin; (D) macmoondongtang, 25 mg/mL.

Fig 2B–2D, the WBC, eosinophil, and neutrophil counts in the dexamethasone treatment group were significantly decreased compared with the OVA-induced asthma group. However, *macmoondongtang* treatment suppressed their proliferation dose-dependently. Especially, the number of neutrophils in the group treated with1500 mg/kg *macmoondongtang* was significantly decreased (Fig 2D) similar to the group treated with dexamethasone.

### *Macmoondongtang* induces dose-dependent suppression of IgE

IgE levels usually surge in allergy and increase significantly in asthma [29]. Serum IgE levels in OVA-induced asthma group (60.17 ± 8.135 ng/mL) were significantly increased compared to the levels (13.98 ± 3.877 ng/mL) in the control (Fig 3 and S1 File). Although the serum IgE level in dexamethasone-treated group was higher than in the control, the increased level of IgE induced by OVA treatment was down-regulated to 34.00 ± 8.351 ng/mL (57% compared with OVA treatment) after dexamethasone treatment. *Macmoondongtang* controlled the IgE level in a dose-dependent manner: 54.91 ± 10.141 ng/mL in the group treated with 150 mg/kg and 44.35 ± 9.618 ng/mL in the group exposed to 1500 mg/kg. There was no significant difference in serum IgE levels between groups treated with dexamethasone and 1500 mg/kg of *macmoondongtang*, which suggested that 1500 mg/kg *macmoondongtang* was adequate to treat asthma patients.

### *Macmoondongtang* dose-dependently inhibits typical morphological changes in OVA-induced asthma model

To evaluate morphological changes in the lung, H&E staining was performed (Fig 4A). The OVA-treated group showed typical morphological changes such as inflammatory cell infiltration (purple color) near bronchioles and vessels, pulmonary hyperplasia (thickening inner bronchiole), and mucous hypersecretion (purple color) in the bronchiole (Fig 4Bb) compared with the control (Fig 4Aa). Dexamethasone controlled OVA-induced morphological changes in the lung by decreasing inflammatory cell infiltration near bronchioles and vessels and inhibiting basal membrane thickening of inner bronchioles and mucous secretion (Fig 4Ac). It Fewer differences in lung morphology were detected between groups treated with OVA (Fig 4Ab) and 150 mg/kg *macmoondongtang* (Fig 4Ad). However, OVA-induced asthmatic

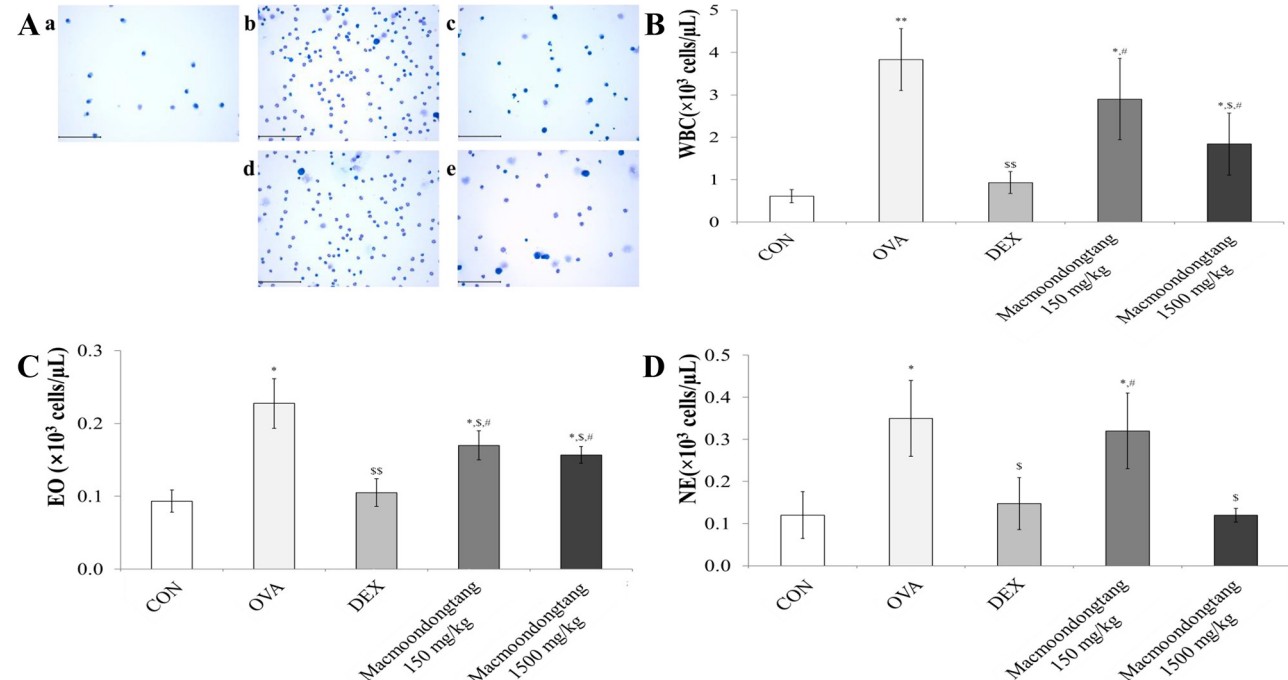

**Fig 2. *Macmoondongtang* treatment suppressed infiltration of both inflammatory cells and immune cells such as white blood cells (WBCs), eosinophils, and neutrophils.** (A) *Macmoondongtang* effectively decreased OVA-induced proliferation of inflammatory cells. (B) *Macmoondongtang* dose-dependently down-regulated the population of WBCs. (C) *Macmoondongtang* effectively suppressed the number of eosinophils. (D) *Macmoondongtang* significantly and dose-dependently decreased neutrophil proliferation. **a**, vehicle control; **b**, asthma induction; **c**, dexamethasone; **d**, 150 mg/kg/day *macmoondongtang*; **e**, 1500 mg/kg/day *macmoondongtang*. Each bar represents the mean ± SD (n = 8). $^*p < 0.05$ vs. control; $^{**}p < 0.001$ vs. control; $^\$p < 0.05$ vs. asthma induction; $^{\$\$}p < 0.01$ vs. asthma induction; $^\#p < 0.05$ vs. dexamethasone. Scale Bar = 100 μm. Magnification, ×400.

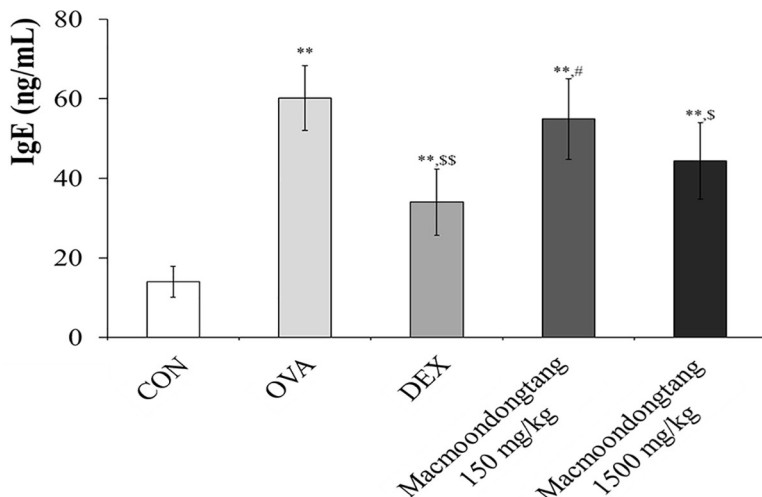

**Fig 3. *Macmoondongtang* treatment effectively suppressed serum IgE level in a dose-dependent manner.** Serum IgE level in the group treated with 1500 mg/kg *macmoondongtang* was compared with that of dexamethasone-treated group. Each bar represents the mean ± SD (n = 16). $^{**}p < 0.001$ vs. control; $^\$p < 0.05$ vs. asthma induction; $^{\$\$}p < 0.01$ vs. asthma induction; $^\#p < 0.05$ vs. dexamethasone.

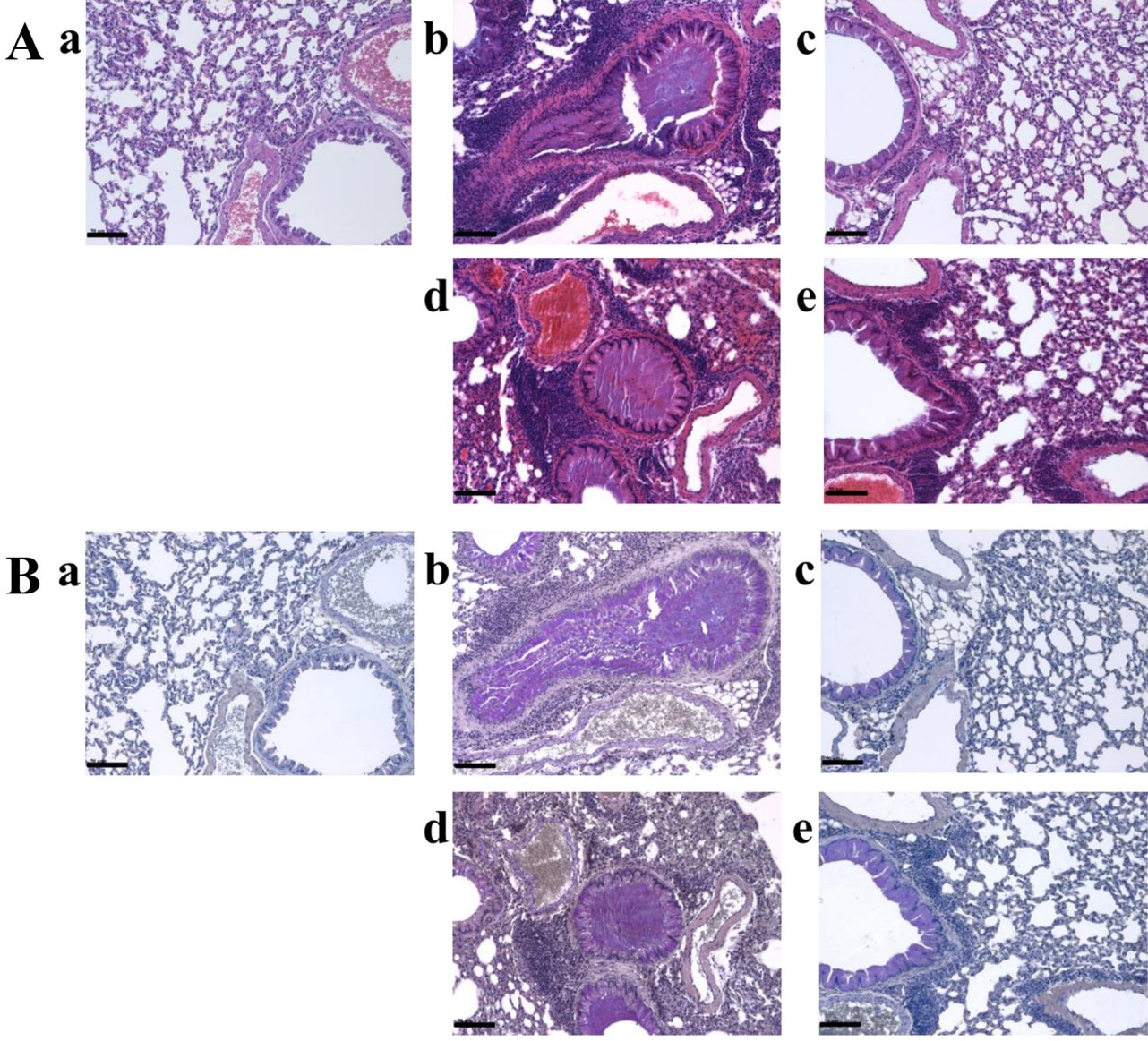

**Fig 4. *Macmoondongtang* dose-dependently inhibited typical morphological changes in the lungs of mice with OVA-induced asthma.** (A) *Macmoondongtang* controlled base membrane thickening caused by epithelial cell hyperplasia, inflammatory infiltration near bronchiole and vessel, mucous hypersecretion, and other changes based on H&E staining. (B) *Macmoondongtang* dose-dependently suppressed mucous secretion based on PAS staining. **a**, vehicle control; **b**, asthma induction; **c**, dexamethasone; **d**,150 mg/kg/day *macmoondongtang*; **e**, 1500 mg/kg/day *macmoondongtang*. Br, bronchioalveolar region; V, vessel. Scale Bar = 50 μm. Magnification, ×200.

changes in the lung were reversed by treatment with 1500 mg/kg *macmoondongtang* (Fig 4Ae), although the recovery was less than that induced by dexamethasone.

In the case of asthma patients, breathing difficulty is usually observed due to factors such as pulmonary base membrane thickening due to epithelial cell hyperplasia and mucous hypersecretion [16]. Mucous hypersecretion in the bronchiole can directly inhibit breathing, and therefore, is a very important parameter for the evaluation of asthma severity. To determine the effect of *macmoondongtang* on mucous secretion, PAS staining was conducted (Fig 4B). In the control, no mucus was detected in the bronchiole (Fig 4Ba). However, in the OVA-treated group (Fig 4Bb) and the group treated with 150 mg/kg macmoondongtang (Fig 4Bd), the

bronchiole was almost obstructed by mucus (purple color). There was little difference in mucus levels between the groups treated with dexamethasone (Fig 4Bc) and 1500 mg/kg *macmoondongtang* (Fig 4Be). These results indicate that *macmoondongtang* dose-dependently suppresses mucous secretion.

### *Macmoondongtang* effectively induces inactivation of both Th1 and Th2 cell transcription factors

Asthma is caused by the imbalance of Th1 and Th2 cells [2]. T-bet is a Th1 cell transcription factor [30, 31] while GATA-3 is a Th2 cell transcription factor [9]. GATA-3 immunofluorescent assay was conducted to evaluate the changes in the activation levels of Th1 cell transcription factor, T-bet, and Th2 cell transcription factor (Fig 5). The activated transcription factors are translocated from the cytoplasm to nuclei. The distribution of T-bet (green color, Fig 5B) was similar to that of GATA-3 (red color, Fig 5C). OVA treatment activated both Th1 and Th2 transcription factors in the nuclei (Fig 5Db). However, in the other groups, both T-bet and GATA-3 remained localized to the cytoplasm (Fig 5Da, 5Dc & 5Dd). These results indicate that *macmoondongtang* inhibited the activation of both Th1 and Th2 cell transcription factors.

### *Macmoondongtang* significantly suppresses both gene and protein expression of IFN-γ and IL-12

To analyze the levels of Th1-related cytokines such as IFN-γ and IL-12, we conducted RT-PCR (Fig 6A and S1 File) and IHC staining (Fig 6B and 6C). As shown in Fig 6A, *macmoondongtang* effectively and dose-dependently inhibited *IFN-γ* gene expression, which was dramatically up-regulated by OVA treatment. *IFN-γ* gene expression in the group treated with 1500 mg/kg *macmoondongtang* was similar to that of the group treated with dexamethasone. Similar to changes in the expression of *IFN-γ* gene, the protein expression induced by OVA treatment was also suppressed by *macmoondongtang* (Fig 6B). Changes in gene and protein expression of IL12p40 were similar to those of IFN-γ (Fig 6A and 6C).

### *Macmoondongtang* controls the expression of IL-4 and IL-5, but not IL-13, in a dose-dependent manner

OVA treatment significantly suppressed the expression of *IL-5* and *IL-13* genes, but not that of *IL-4* gene, although it varied depending on treatment materials such as sterilized tap water (control), dexamethasone, and *macmoondongtang* (Fig 7A and S1 File). However, *macmoondongtang* significantly down-regulated the expression of *IL-5* gene alone. The gene expression of Th2-related cytokines differed from their protein levels in the respiratory system (Fig 7B–7D). Although the *IL-5* gene expression was statistically significant after *macmoondongtang* treatment, the protein expression of all Th2-related cytokines in the present study such as IL-4, IL-5, and IL-13 was dose-dependently suppressed by *macmoondongtang* treatment. The levels of IL-4 (Fig 7Be), IL-5 (Fig 7Ce) and IL-13 proteins (Fig 7De) in the group treated with 1500 mg/kg *macmoondongtang* were significantly decreased compared with those of the dexamethasone-treated group (Fig 7Cc & 7Dc).

### *Macmoondongtang* modulates the expression of TNF-α, but not IL-6

OVA treatment increased the levels of only *TNF-α* gene (Fig 8A and S1 File). Dexamethasone significantly decreased the expression of *TNF-α* gene while *macmoondongtang* dose-dependently suppressed its level. Especially, treatment with 1500 mg/kg *macmoondongtang* suppressed the expression of *TNF-α* gene similar to that of dexamethasone-treated group.

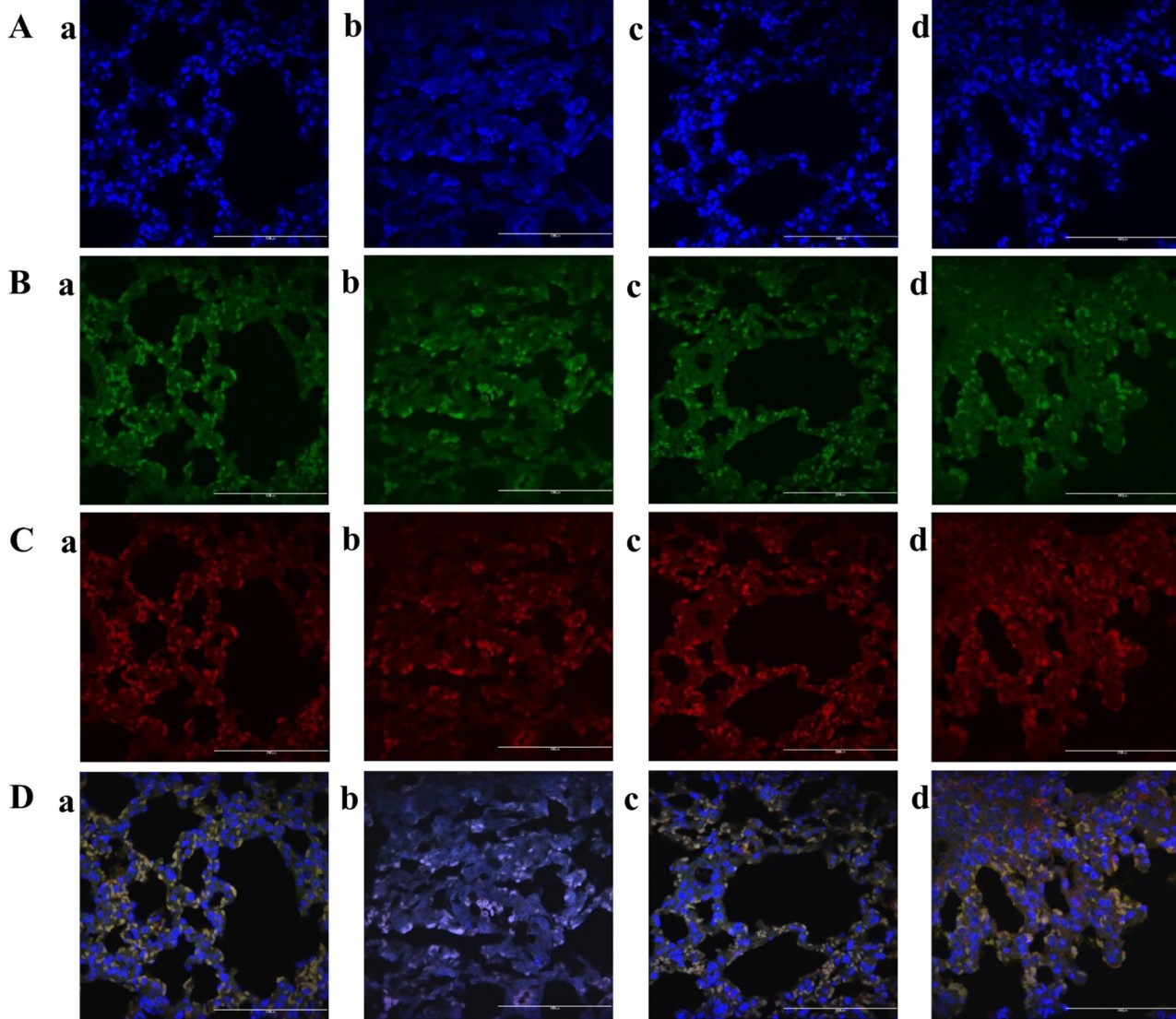

**Fig 5. *Macmoondongtang* effectively induced the inactivation of both Th1 and Th2 cell transcription factors.** (A) Nuclei are shown in blue color by DAPI staining. (B) Images displaying results of T-bet staining by FITC. (C) GATA-3 staining by AlexaFluor® 555. (D) Merged images of nuclei stained with DAPI, T-bet stained with FITC, and GATA-3 stained with Alexa Fluor® 555. Treatment with dexamethasone and *macmoondongtang* completely abrogated the activation of both Th1 cell transcription factor T-bet and Th2 cell transcription factor GATA-3. **a**, vehicle control; **b**, asthma induction; **c**, dexamethasone; **d**,1500 mg/kg/day macmoondongtang. Scale Bar = 100 μm. Magnification, ×1000.

*Macmoondongtang* down-regulated TNF-α expression in a dose-dependent manner (Fig 8B). However, it did not suppress the level of IL-6, which was increased by OVA treatment (Fig 8Cc, 8Cd & 8Ce). The TNF-α level was decreased by treatment with 1500 mg/kg *macmoondongtang* (Fig 8Be), similar to that of dexamethasone treatment (Fig 8Bc).

## Discussion

Environmental allergens inducing asthma are classified into two types: indoor allergens (for e.g., pet dander, dust mite, tobacco smoke, etc.) and outdoor allergens (for e.g., pollen, environmental pollutants, cold temperature, etc.) [32]. At first, antigen-presenting cells (APCs) recognize repeated invasion by allergens, leading to asthma. Studies have been performed to

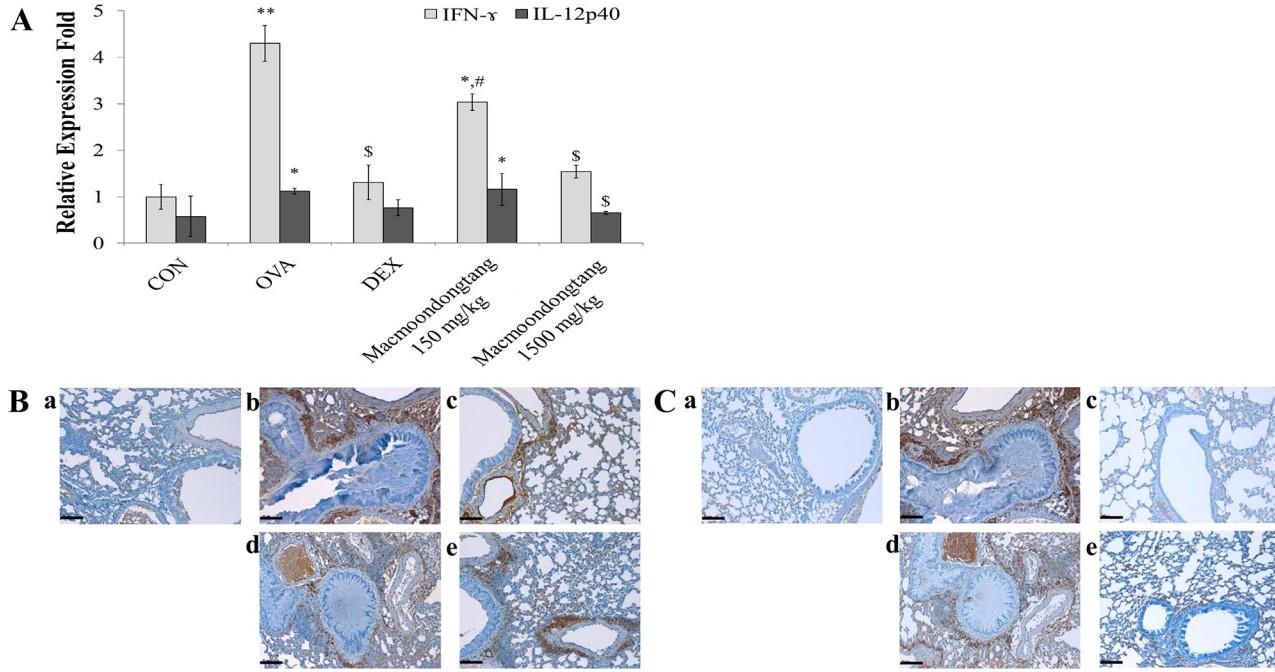

**Fig 6. Treatment with *macmoondongtang* significantly down-regulated gene and protein expression of IFN-γ and IL-12p40.** *Macmoondongtang* dose-dependently controlled both *IFN-γ* gene expression quantified by RT-PCR (A) and IFN-γ protein levels assessed by IHC (B). *Macmoondongtang* also regulated *IL-12* gene expression and up-regulated gene and protein expression of IL-12p40 in the lung (C). **a**, vehicle control; **b**, asthma induction; **c**, dexamethasone; **d**, 150 mg/kg/day *macmoondongtang*; **e**, 1500 mg/kg/day *macmoondongtang*. Each bar represents the mean ± SD (n = 8). $^{*}p < 0.05$ vs. control; $^{**}p < 0.001$ vs. control; $^{\$}p < 0.05$ vs. asthma induction; $^{\#}p < 0.05$ vs. dexamethasone. Scale Bar = 50 μm. Magnification, ×200.

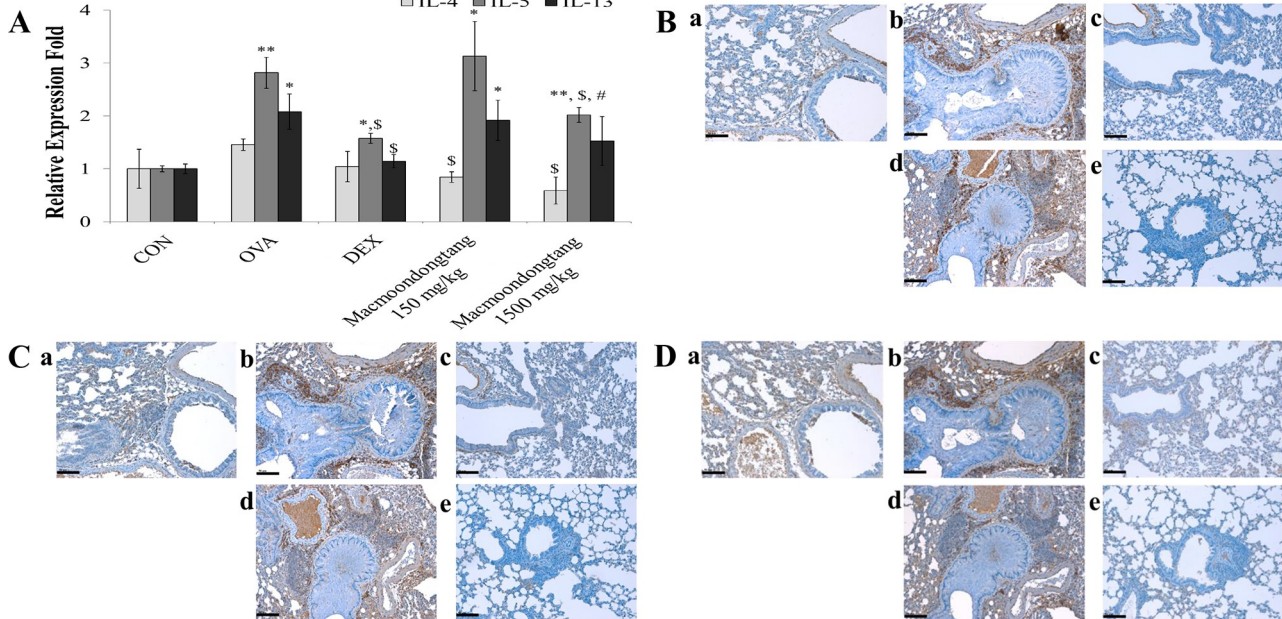

**Fig 7. *Macmoondongtang* dose-dependently suppressed Th2-related cytokines such as IL-4, IL-5, and IL-13.** *Macmoondongtang* significantly and effectively inhibited the gene expression of *IL-5* (A) and IL-5 protein level (C). The treatment dose-dependently controlled the IL-4 protein expression (B). Similar changes at the gene (A) and protein levels of IL-13 (D) were observed as those for IL-5. **a**, vehicle control; **b**, asthma induction; **c**, dexamethasone; **d**, 150 mg/kg/day *macmoondongtang*; **e**, 1500 mg/kg/day *macmoondongtang*. Each bar represents the mean ± SD (n = 8). $^{*}p < 0.05$ vs. control; $^{**}p < 0.001$ vs. control; $^{\$}p < 0.05$ vs. asthma induction; $^{\#}p < 0.05$ vs. dexamethasone. Scale Bar = 50 μm. Magnification, ×200.

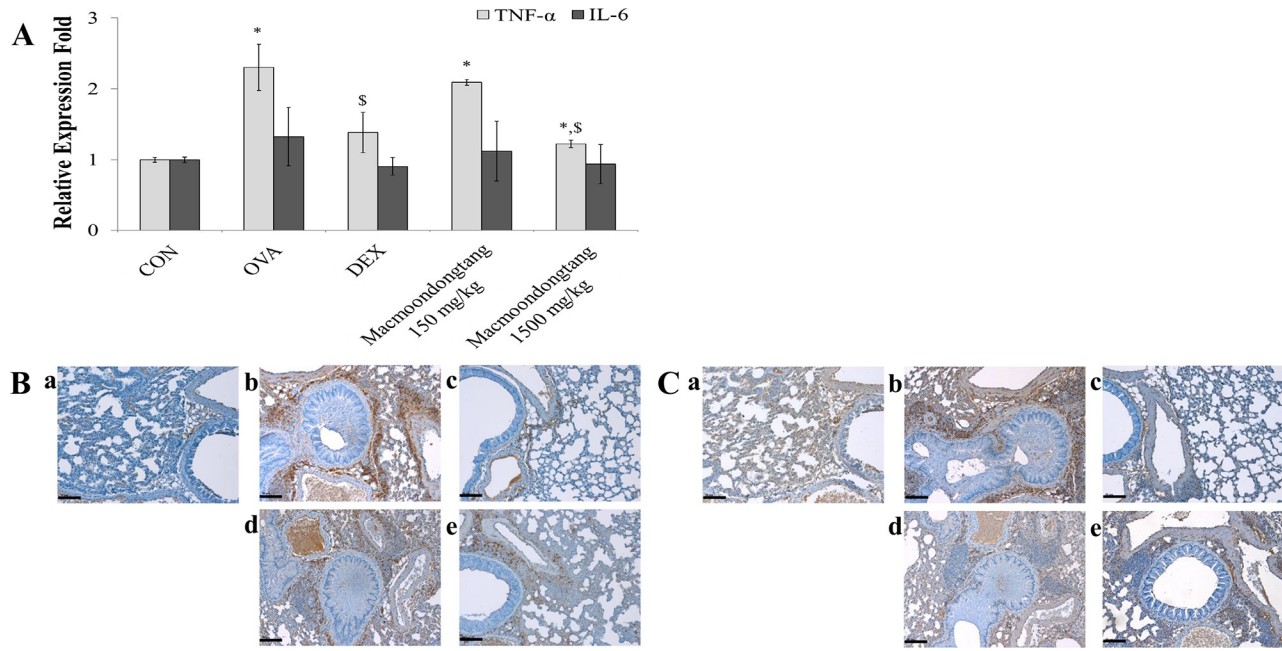

**Fig 8.** *Macmoondongtang* **treatment significantly inhibited** *TNF-α* **gene and protein expression, but not that of IL-6.** *Macmoondongtang* dose-dependently suppressed the gene (A) and protein expression (B) of TNF-α but not that of IL-6 expression (C). **a**, vehicle control; **b**, asthma induction; **c**, dexamethasone; **d**,150 mg/kg/day *macmoondongtang*; **e**, 1500 mg/kg/day *macmoondongtang*. Each bar represents the mean ± SD (n = 8). $^*p < 0.05$ vs. control; $^\$ p < 0.05$ vs. asthma induction. Scale Bar = 50 μm. Magnification, ×200.

verify the relation between type 1 allergy and IgE, an important and representative biomarker [29, 33]. Accordingly, IgE increase is a very important factor that triggers asthma, suggesting the need to decrease IgE expression induced by OVA exposure. Immune response is critical to maintain the homeostasis of Th1 and Th2 cells in living organisms. The hypothesis of Th1/Th2 balance for immune system was introduced in the 1980s [34]. Asthma is one of the diseases triggered by Th1/Th2 imbalance and according to the theory [35], the balance between Th1 cells and Th2 cells is generally maintained. However, an up-regulation of Th2 cells including related cytokines may trigger asthma. IFN-γ and IL-12 are Th1 cytokines, whereas IL-4, IL-5, and IL-13 are Th2 cytokines, and IL-6 and TNF-α are Th17-related cytokines[10].

T-bet is a factor associated with Th1 cell transcription. T-bet and IFN-γ show a positive feed-back [30, 36]. IL-12p40 is one of the key factors regulating Th1-related cytokines in that it specifically stimulates IFN-γ production in asthma [5]. As shown in Fig 6, changes in IFN-γ gene and protein levels were very similar to those of IL-12p40 while *macmoondongtang* effectively inhibited their increases.

In asthma patients, the Th2-related cytokines such as IL-4, IL-5, and IL-13 are usually increased [37–39]. GATA-3 is a Th2 cell transcription factor. IL-4 increases its activation. Consequentially, the regulation of GATA-3 and IL-4 produces Th2-related cytokines [9]. *Macmoondongtang* not only significantly inactivated both Th1 and Th2 transcription factors (Fig 5), but also suppressed gene and protein levels of IFN-γ and IL-4 directly linked to each transcription factor (Figs 6A, 6B, 7A and 7B).

Recognition of APCs by Th2 cells not only releases cytokines such as IL-4, IL-5, and IL-13 that stimulate IgE surge via B cell activation, but also increases the eosinophil population induced by IL-5 to initiate airway remodeling [33, 40]. In the lung, increased IL-13 expression is extensively associated with airway hyper-responsiveness such as inflammation, mucous hypersecretion, epithelial fibrosis, and airway obstruction [16]. *Macmoondongtang* controlled

the levels of all Th2-related cytokines investigated such as IL-4, IL-5, and IL-13. It specifically suppressed the expression of IL-4 and IL-5 (Fig 7A–7C). These results explain the anti-asthmatic effect of *macmoondongtang* mediated via suppression of serum IgE level (Fig 3) to reverse the typical morphological changes associated with asthma such as epithelial hyperplasia, mucous hypersecretion, and airway obstruction (Fig 4).

Th17-related cytokines such as TNF-α, IL-6, and IL-1β are important in the control of asthma onset [41]. TNF-α is one of the chemo-attractants for neutrophils and eosinophils [18]. It induces death of eosinophils via oxidative stress in pulmonary epithelial cells [42]. Although eosinophil numbers are increases in many asthma patients, other patients show a surge in neutrophil levels [43]. In the present study, *macmoondongtang* effectively suppressed the proliferation of eosinophils and neutrophils in BALF and inhibited eosinophil infiltration near bronchioles and alveoli, and vessels via TNF-α modulation (Fig 2). The expression of IL-6 in asthma patients is significantly increased [44]. The IL-6 induced by allergens stimulates the release of IL-4 and IL-13 from Th2 cells [12]. IL-6 may be one of the key factors maintaining the balance of helper T cells to control Treg cells or Th17 cells [44].

Several reports suggest the anti-asthmatic effects of herbal extracts [45–47] or compounds derived from natural products [48]. However, the bioactivity of may be associated with specific compounds derived from herbs or mushrooms. The anti-asthmatic effect was attributed to bioactive ingredient(s) in the natural product. *Macmoondongtang* used in this study was obtained as an extract from several herbs such as *Glycyrrhizae radix*, *Oryza sativa*, *Zizyphus jujube* Inermis, *Ophiopogon japonicas*, *Pinellia ternate*, and *Panax ginseng*. A review of literature suggests that *macmoondongtang* contains anti-asthmatic markers such as glycyrrhizin and anti-inflammatory markers such as liquiritin. To determine the bioactive compounds in macmoondongtang, HPLC analysis was conducted. Results showed that glycyrrhizin (0.38%) and liquiritin (0.02%) were present in the *macmoondongtang* extract. Glycyrrhizin is one of the major compounds in *Glycyrrhiza uralensis* and *Glycyrrhiza glabra*. Ram *et al*. (2006) reported that glycyrrhizin exhibits an immunomodulatory effect in a mouse model of asthma [49]. Oral administration of glycyrrhizin (2.5 to 20 mg/kg) to OVA-induced mice resulted in a suppression of IL-4, IL-5, IFN-γ, and IgE levels. In the present study, we identified liquiritin content in *macmoondongtang* extract. Liquiritin modulates inflammation via inhibition of pro-inflammatory mediators such as inducible nitric oxide synthase, cyclooxygenase (COX)-2, TNF-α, IL-1β, and IL-6 [50, 51]. Thus, liquiritin occurring in *macmoondongtang* may alleviate the inflammatory symptoms caused by asthma. Although *G. auralensis* contains glycyrrhizin and *G. glabra* has liquiritin, each herb exhibits a typical anti-asthmatic effect.

Ram *et al*. (2006) have reported that an oral dose starting with 5 mg/kg of liquiritin was effective in mice [49]. In the present study, we found that *macmoondongtang* resulted in an anti-asthmatic effect at a dose of 1500 mg/kg in our animal model. The dose of 1500 mg/kg corresponds to 5.7 mg/kg of glycyrrhizin. Thus, the curative dose in the present study was consistent with that of the previous report, suggesting that glycyrrhizin contained in *macmoondongtang* is a possible therapeutic candidate for the management of asthma. Furthermore, we evaluated a daily oral dose of 7.3 g *macmoondongtang* for treatment of asthma. The therapeutically effective oral dose of *macmoondongtang* in mouse was 1500 mg/kg/day. The conversion factor between human and mouse is 12.3 [21]. Therefore, if the effective dose of *macmoondongtang* in mice is 1500 mg/kg/day, the human equivalent dose is 7300 mg/60kg/day. The recommended daily intake of *macmoondongtang* for human is 9 g. Thus, the optimal dose for asthma treatment is less than its daily intake and then we could confirm that oral intake of macmoondongtang at a dose of 7.3 g might prevent the occurrence of asthma.

From the results we concluded that Macmoondongtang treatment alleviates asthma symptoms and modulate the Th1-/Th2- related cytokines. Glycyrrhizin and liquiritin could be the major the active therapeutic components.

## Supporting information

**S1 File. This file is raw data for making graphs in Fig 2, Fig 3, Fig 6, Fig 7 and Fig 8.** (XLSX)

## Author Contributions

**Conceptualization:** Seung Sik Cho, Dae-Hun Park.

**Data curation:** Seung Sik Cho, Dae-Hun Park.

**Formal analysis:** Soon-Young Lee, Bossng Kang, So-Hyeon Bok.

**Funding acquisition:** Seung Sik Cho, Dae-Hun Park.

**Investigation:** Soon-Young Lee, Bossng Kang, So-Hyeon Bok, Dae-Hun Park.

**Methodology:** Soon-Young Lee, Bossng Kang, So-Hyeon Bok.

**Project administration:** Dae-Hun Park.

**Resources:** Dae-Hun Park.

**Supervision:** Dae-Hun Park.

**Visualization:** Dae-Hun Park.

**Writing – original draft:** Soon-Young Lee.

**Writing – review & editing:** Bossng Kang, So-Hyeon Bok, Seung Sik Cho, Dae-Hun Park.

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
