## [Decision Letter · Decision Letter 0]

24 Jul 2019

PONE-D-19-17023

Macmoondongtang modulates asthma-related changes via modulation of TNF-a and T-bet & IFN-r in an asthma murine model

PLOS ONE

Dear Professor Park,

Thank you for submitting your manuscript to PLOS ONE. After careful consideration, we feel that it has merit but does not fully meet PLOS ONE’s publication criteria as it currently stands. Therefore, we invite you to submit a revised version of the manuscript that addresses the points raised during the review process.

We would appreciate receiving your revised manuscript by Sep 07 2019 11:59PM. To enhance the reproducibility of your results, we recommend that if applicable you deposit your laboratory protocols in protocols.io, where a protocol can be assigned its own identifier (DOI) such that it can be cited independently in the future. For instructions see: http://journals.plos.org/plosone/s/submission-guidelines#loc-laboratory-protocols

We look forward to receiving your revised manuscript.

Kind regards,

Michal A Olszewski, DVM, PhD

Academic Editor

PLOS ONE

Journal Requirements:

Additional Editor Comments:

While parts of the manuscript were found to be interesting, there were several serious concerns that need to be addressed before the manuscript could meat the criteria for publication in PLOS One.

Please ensure that all the reviewers' comments are addressed and the conclusions always match the presented data. Please make sure that the N values for individual groups/time points are specified and they type of statistical analysis is always clearly disclosed.

Finally, in addition to the reviewers' comments, the histology images are low resolution and often out of focus, which makes the evaluation of histology panels difficult.

Please retake the pictures as needed and provide the high-resolution files to ensure that the histology data are clearly presented. Whenever, possible the multi-panel figures could be divided into multiple figures so that the individual histology images could be magnified for improved visibility.

Reviewers' comments:

Reviewer's Responses to Questions

**Comments to the Author**

1. Is the manuscript technically sound, and do the data support the conclusions?

Reviewer #1: No

Reviewer #2: No

2. Has the statistical analysis been performed appropriately and rigorously? 

Reviewer #1: No

Reviewer #2: No

3. Have the authors made all data underlying the findings in their manuscript fully available?

Reviewer #1: Yes

Reviewer #2: Yes

4. Is the manuscript presented in an intelligible fashion and written in standard English?

Reviewer #1: No

Reviewer #2: No

5. Review Comments to the Author

Reviewer #1: Comment #1

The manuscript entitled with “Macmoondongtang modulates asthma-related changes via modulation of TNF-a and T-bet & IFN-r in an asthma murine model” brought some interesting findings. However, it is not convincible for its description and conclusion with several reasons as follows:

1) Macmoondongtang has been used as a traditional prescription for pulmonary disease in Korea. However, there is no report about the mechanism involved in its effect. In the present study, the authors described how Macmoondongtang as a repiratory medicine has effect on Ashma in a murine model. Since Macmoondongtang is a traditional prescription for pulmonary disease, surprisingly, there is no any information on how Macmoondongtang has been clinically used in Ashma treatment at all. So, it is really weird for me to choose Ashma instead of the other disease models for study on Macmoondongtang.

2) According to the results, it looks like that Macmoondongtang play role in modulation of expression of Th1, Th2 and Th17-related cytokines. Why did the manuscript choose a tittle as modulation of TNF-a and T-bet & IFN-r? It doesn't make any sense to me.

3) The results showed that Macmoondongtang contained glycyrrhizin and liquiritin. Theoretically glycyrrhizin and liquiritin could have some role in modulation of the cytokine expression. But how did you exclude effects of some unknown components in Macmoondongtang.

Comment #2

Has the statistical analysis been performed appropriately and rigorously? No Although the manuscript showed thirty-fine mice were divided into five groups, it was not clear of how many mice had being used in each figure. So, it is very confused about how they calculated the data without showing “n” in each figure.

Comment #3

Is the manuscript presented in an intelligible fashion and written in standard English? No

There are a lot of places to be rewrote in a standard English in this manuscript, for example, ”It is important to decrease IgE expression increased by ovalbumin treatment”, “T-bet is a factor for Th1 cell transcription…., to the end of this paraph”, and so on.

Some minor comments:

1) For material and methods, there were a lot of description to be improved, for example, which samples were used for RNA extraction? how the samples were prepared for IFA? and so on.

2) What’s magnitude for those microscopic figures? And the quality of those microscopic figures should be also improved.

Reviewer #2: In this study, Lee et al. reported that macmoondongtang, a traditional prescription for treating respiratory disease, could modulate host immune responses during OVA-induced asthma in a mice model. They reported that macmoondongtang treatment could reduce serum IgE level, mucus hypersecretion, inflammatory cell infiltration in the lungs. They also reported that macmoondongtang treatment leads to reduced expression of Th1/Th2 transcriptional factors as well as related cytokines such as IFN-g, IL-12, IL-4 and IL-5, IL-13 on mRNA level or protein level. The data is interesting; however, the manuscript has severe limitations that the authors need to deal with before publications.

1. The written Englished should be revised significantly. There are many grammar mistakes in the manuscript.

2. The conclusions are overstated and need revision. For example,

a. it is not convincing that glycyrrhizin and liquiritin were the effective ingredients of macmoondongtang.

b. There is no data supporting the title: “Macmoondongtang modulates asthma-related changes via modulation of TNF-a and Tbet & IFN-g in an asthma murine model”. They showed correlation, nut not causality.

3. The reviewer also concerned about the quality of some data. For example, fig 5, it is hard to tell the localization of the transcriptional factors (Tbet and GATA3) at the current magnification. Fig 6, 7, 8, it would be nice to quantify the IHC signals.

4. Methods are missing: RNA isolation, cDNA synthesis, qPCR methods, Elisa for IgE,

5. Fig 1, please also include the control HPLC data for glycyrrhizin and liquiritin.

6. For statistical analysis, how many times have these experiments been repeated? It says mean ± SEM in the figure legends but mean ± SD in the Materias and Methods, which one is correct?

6. PLOS authors have the option to publish the peer review history of their article (what does this mean?). If published, this will include your full peer review and any attached files.

Reviewer #1: No

Reviewer #2: No

---

## [Author Response · Author response to Decision Letter 0]

19 Aug 2019

Reviewer #1

Comment #1

The manuscript entitled with “Macmoondongtang modulates asthma-related changes via modulation of TNF-a and T-bet & IFN-r in an asthma murine model” brought some interesting findings. However, it is not convincible for its description and conclusion with several reasons as follows:

1) Macmoondongtang has been used as a traditional prescription for pulmonary disease in Korea. However, there is no report about the mechanism involved in its effect. In the present study, the authors described how Macmoondongtang as a repiratory medicine has effect on Ashma in a murine model. Since Macmoondongtang is a traditional prescription for pulmonary disease, surprisingly, there is no any information on how Macmoondongtang has been clinically used in Ashma treatment at all. So, it is really weird for me to choose Ashma instead of the other disease models for study on Macmoondongtang.

ans) Thank you for your kind advise.

Macmoondongtang has been used as oriental medicine in Korea and China and is still commercialized. Macmoondongtang was generally used to cure bronchitis, asthma, pneumonia, cough, and sputum production. The mechanism of anti-asthmatic activity of Macmoondongtang has not been reported. Therefore, we have demonstrated the anti-asthmatic effect of Macmoondongtangin animal experiments used commercial products (see Materials and methods). The effective dose of Macmoondongtang in animal studies was determined to be 1500 mg/kg. We calculated the dose of human based on the results of animal experiments. Human dose was included in the daily intake range of commercial products (see, discussion section). The important facts in this manuscript are that anti-asthmatic effect has been demonstrated within the stable dose range of Macmoondongtang, and the action mechanism has also been clearly identified.

2) According to the results, it looks like that Macmoondongtang play role in modulation of expression of Th1, Th2 and Th17-related cytokines. Why did the manuscript choose a tittle as modulation of TNF-a and T-bet & IFN-r? It doesn't make any sense to me.

Ans) Thank you so much for your comments and I agreed with your comments. I think that the manuscript title should be changed to “Macmoondongtang controls the asthmatic changes via modulation of Th1-/Th2-/Th17-related factors in an asthma murine model”.

3) The results showed that Macmoondongtang contained glycyrrhizin and liquiritin. Theoretically glycyrrhizin and liquiritin could have some role in modulation of the cytokine expression. But how did you exclude effects of some unknown components in Macmoondongtang.

ans) Thank you for your kind advise. The effective dose of glycyrrhizin was reported to be 2.5-20 mg/kg in the mouse model (Ram et al) and liquiritin has been reported to be efficacious at a minimum dose of 5 mg/kg in mouse model (Ram et al). We reported that in the discussion section that Macmoondongtang contained 0.38% glycyrrhizin. Glycyrrhizin in Macmoondongtang was calculated based on the content and correlated with animal experimental results. The dose of 1500 mg/kg Macmoondongtang is equivalent to 5.7 mg/kg glycyrrhizin. These results are correlated with previous reports (efficacy range 2.5 - 20 mg/kg). It can be interpreted that glycyrrhizin present in Macmoondongtang shows the main anti-asthmatic effect. Based on the content of liquiritin, the effective dose (5 mg/kg or more) is insufficient. Therefore, liquiritin was thought to be a cofactor for anti-asthmatic effect.

Comment #2

Has the statistical analysis been performed appropriately and rigorously? No Although the manuscript showed thirty-fine mice were divided into five groups, it was not clear of how many mice had being used in each figure. So, it is very confused about how they calculated the data without showing “n” in each figure.

Ans) Thank you so much for your advices. This study was conducted twice time using with 70 heads, for one study 35 mice had been used and using with same method the other study was done. I added this method in the Materials and Methods section and used numbers of animals for each graph was described in the Figure Legends. 

Comment #3

Is the manuscript presented in an intelligible fashion and written in standard English? No

There are a lot of places to be rewrote in a standard English in this manuscript, for example, ”It is important to decrease IgE expression increased by ovalbumin treatment”, “T-bet is a factor for Th1 cell transcription…., to the end of this paraph”, and so on.

Ans) Thank you so much for your advice. Although this manuscript had been edited by professional editing company (I attached the certificate file of editing) revised manuscript would be done by editing company again.

Some minor comments:

1)For material and methods, there were a lot of description to be improved, for example, which samples were used for RNA extraction? how the samples were prepared for IFA? and so on.

Ans) Thank you so much for your comments and I add the description about materials and methods in the Section. 

2) What’s magnitude for those microscopic figures? And the quality of those microscopic figures should be also improved.

Ans) Thank you so much for your advice and I added the magnification in each photo. For improving the photos’ quality, I adjusted them again.

Reviewer #2

In this study, Lee et al. reported that macmoondongtang, a traditional prescription for treating respiratory disease, could modulate host immune responses during OVA-induced asthma in a mice model. They reported that macmoondongtang treatment could reduce serum IgE level, mucus hypersecretion, inflammatory cell infiltration in the lungs. They also reported that macmoondongtang treatment leads to reduced expression of Th1/Th2 transcriptional factors as well as related cytokines such as IFN-g, IL-12, IL-4 and IL-5, IL-13 on mRNA level or protein level. The data is interesting; however, the manuscript has severe limitations that the authors need to deal with before publications.

1. The written Englished should be revised significantly. There are many grammar mistakes in the manuscript.

Ans) Thank you so much for your advice. Although this manuscript had been edited by professional editing company (I attached the certificate file of editing) revised manuscript would be done by editing company again.

2. The conclusions are overstated and need revision. For example,

a. it is not convincing that glycyrrhizin and liquiritin were the effective ingredients of macmoondongtang.

ans) Thanks for your kind advise. Macmoondongtang has been used as oriental medicine in Korea and China and is still commercialized. Macmoondongtang was generally used to cure bronchitis, asthma, pneumonia, cough, and sputum production. The mechanism of anti-asthmatic activity of Macmoondongtang has not been reported. Therefore, we have demonstrated the anti-asthmatic effect of Macmoondongtangin animal experiments used commercial products (see Materials and methods). The effective dose of Macmoondongtang in animal studies was determined to be 1500 mg/kg. We calculated the dose of human based on the results of animal experiments. Human dose was included in the daily intake range of commercial products (see, discussion section). The important facts in this manuscript are that anti-asthmatic effect has been demonstrated within the stable dose range of Macmoondongtang, and the action mechanism has also been clearly identified.

liquiritin and glycyrrhizin were designated as the main marker of Macmoondongtang and the test method for content analysis 

(see. https://oasis.kiom.re.kr/oasis/pres/prdetailView2.jsp?idx=32&selectname=null&srch_menu_nix=null#view04) was defined in KFDA (Korea Food & Drug Administration). In this study, we confirmed that liquiritin and glycyrrhizin are markers for quality control and one of the effective markers.

b. There is no data supporting the title: “Macmoondongtang modulates asthma-related changes via modulation of TNF-a andTbet& IFN-g in an asthma murine model”. They showed correlation, nut not causality.

Ans) Thank you so much for your comments and I agreed with your comments. I think that the manuscript title should be changed to “Macmoondongtang controls the asthmatic changes via modulation of Th1-/Th2-/Th17-related factors in an asthma murine model”

3. The reviewer also concerned about the quality of some data. For example, fig 5, it is hard to tell the localization of the transcriptional factors (Tbet and GATA3) at the current magnification. Fig 6, 7, 8, it would be nice to quantify the IHC signals.

Ans) Thank you so much for your comments. We focused the localization of specific protein in the lung tissue such as in nucleus or in cytoplasm and I evaluated the localization of them based on the criteria that in dissimilar to general color when the different colors of light exist are summed-up they change the white color. Therefore, in the asthma-induction group we could evaluated that both of T-bet (green color) and GATA-3 (red color) existed in the nucleus (blue color) as in the nucleus the white spots were found compared to the other groups. 

4. Methods are missing: RNA isolation, cDNA synthesis, qPCR methods, Elisa for IgE,

Ans) Thank you so much for your comments and I add the description about materials and methods in the Section.

5. Fig 1, please also include the control HPLC data for glycyrrhizin and liquiritin

Ans) Thank you for your kind advise. we modified the figure 1 according to reviewers’ recommendation.

6. For statistical analysis, how many times have these experiments been repeated? It says mean ± SEM in the figure legends but mean ± SD in the Materias and Methods, which one is correct?

Ans) Thank you so much for your comment and there were typos. I amended the expression as “mean ± SD”.

---

## [Decision Letter · Decision Letter 1]

6 Sep 2019

PONE-D-19-17023R1

Macmoondongtang alleviates asthma via modulation of Th1-/Th2-/Th17-related cytokines in a murine model of asthma

PLOS ONE

Dear Professor Park,

Thank you for submitting your manuscript to PLOS ONE. After careful consideration, we feel that it has merit but does not fully meet PLOS ONE’s publication criteria as it currently stands. Therefore, we invite you to submit a revised version of the manuscript that addresses the points raised during the review process.

While the manuscript has improved, both reviewers remain concerned about the over-interpretation of findings/speculative conclusions not fully supported by the data. In order to comply with PLOS publication policy the conclusions need to fully support by the data presented. Although, your data identified a link between the therapeutic effects of Macmoondongtang and certain group of cytokine responses, but they are insufficient to proof that it is the mechanism of drug action. Thus, the conclusions and title of the manuscript need to be adjusted as suggested by the Reviewer 2.

We would appreciate receiving your revised manuscript by Oct 21 2019 11:59PM. To enhance the reproducibility of your results, we recommend that if applicable you deposit your laboratory protocols in protocols.io, where a protocol can be assigned its own identifier (DOI) such that it can be cited independently in the future. For instructions see: http://journals.plos.org/plosone/s/submission-guidelines#loc-laboratory-protocols

We look forward to receiving your revised manuscript.

Kind regards,

Michal A Olszewski, DVM, PhD

Academic Editor

PLOS ONE

Reviewers' comments:

Reviewer's Responses to Questions

**Comments to the Author**

1. If the authors have adequately addressed your comments raised in a previous round of review and you feel that this manuscript is now acceptable for publication, you may indicate that here to bypass the “Comments to the Author” section, enter your conflict of interest statement in the “Confidential to Editor” section, and submit your "Accept" recommendation.

Reviewer #1: (No Response)

Reviewer #2: (No Response)

2. Is the manuscript technically sound, and do the data support the conclusions?

Reviewer #1: Partly

Reviewer #2: Partly

3. Has the statistical analysis been performed appropriately and rigorously? 

Reviewer #1: Yes

Reviewer #2: Yes

4. Have the authors made all data underlying the findings in their manuscript fully available?

Reviewer #1: Yes

Reviewer #2: Yes

5. Is the manuscript presented in an intelligible fashion and written in standard English?

Reviewer #1: No

Reviewer #2: Yes

6. Review Comments to the Author

Reviewer #1: Major comments:

After revise, the manuscript has been improved by the authors. But two major things still remain to be improved, as follows:

1, the writing must be improved. 1) introduction, special paragraph 2 and 3, is not so clear to this study; 2) discussion, which has been improved is not enough. Furthermore, some description is misleading, such as the synergistic role between glycyrrhizin and liquiritin unless there are some experimental finding available.

2, the microscropic figures are in a low quality. My previous comment has mentioned this problem. The authors' response was that it had been adjusted. But there are not clear, I don't know why.

Minor comments:

1, there are still some grammar mistakes to be checked.

2, ref #8, is not completed.

Reviewer #2: In this revised manuscript, the authors report that macmoondongtang could regulate host immune response in a murine model of asthma. Specifically, mice treated with macmoondongtang show decreased neutrophil count in BALF, reduced T-bet, IFN-g, and TNF-a, IL-12, IL-4, and IL-5 genes expression in the lung. While most of my previous concerns have been addressed, the reviewer still has some minor comments about several conclusions in the manuscript. The authors provide no data supports that modulation of Th1/Th2 cytokines is the mechanism of how Macmoondongtang alleviates asthma.

1. I would suggest change the title to “Macmoondongtang modulates Th1-/Th2-related cytokines and alleviates asthma in a murine model”.

2. I would suggest modify the conclusions to: “Macmoondongtang treatment alleviates asthma symptoms and modulate the Th1-/Th2- related cytokines. Glycyrrhizin and liquiritin could be the major the active therapeutic components”.

7. PLOS authors have the option to publish the peer review history of their article (what does this mean?). If published, this will include your full peer review and any attached files.

Reviewer #1: No

Reviewer #2: No

---

## [Author Response · Author response to Decision Letter 1]

17 Sep 2019

Reviewer #1: Major comments:

After revise, the manuscript has been improved by the authors. But two major things still remain to be improved, as follows:

1, the writing must be improved. 1) introduction, special paragraph 2 and 3, is not so clear to this study; 2) discussion, which has been improved is not enough. Furthermore, some description is misleading, such as the synergistic role between glycyrrhizin and liquiritin unless there are some experimental finding available.

Ans) Thank you so much for your comments and I amended several parts depended on your comments. As I agreed with your suggestions and omitted the sentence about the synergistic effects of glycyrrhizin and liquiritin. 

2, the microscropic figures are in a low quality. My previous comment has mentioned this problem. The authors' response was that it had been adjusted. But there are not clear, I don't know why.

Ans) Thank you so much for your comments and all of figures were prepared as tiff files. Although all of them are made which met on the publication criteria during uploading the manuscript they might be unclearly changed. Please understand my situation.

Minor comments:

1, there are still some grammar mistakes to be checked.

Ans) The manuscript was edited by editing company and I attached the certificate of edit.

2, ref #8, is not completed.

Ans) Thank you so much for your comment and I amended the reference #8.

Reviewer #2: In this revised manuscript, the authors report that macmoondongtang could regulate host immune response in a murine model of asthma. Specifically, mice treated with macmoondongtang show decreased neutrophil count in BALF, reduced T-bet, IFN-g, and TNF-a, IL-12, IL-4, and IL-5 genes expression in the lung. While most of my previous concerns have been addressed, the reviewer still has some minor comments about several conclusions in the manuscript. The authors provide no data supports that modulation of Th1/Th2 cytokines is the mechanism of how Macmoondongtang alleviates asthma.

1. I would suggest change the title to “Macmoondongtang modulates Th1-/Th2-related cytokines and alleviates asthma in a murine model”.

Ans) Thank you so much for your suggestion and I amended the manuscript’s title.

2. I would suggest modify the conclusions to: “Macmoondongtang treatment alleviates asthma symptoms and modulate the Th1-/Th2- related cytokines. Glycyrrhizin and liquiritin could be the major the active therapeutic components”.

 Ans) Thank you so much for your suggestion and I amended the manuscript’s title.

---

## [Decision Letter · Decision Letter 2]

8 Oct 2019

PONE-D-19-17023R2

Macmoondongtang modulates Th1-/Th2-related cytokines and alleviates asthma in a murine model

PLOS ONE

Dear Professor Park,

Thank you for submitting your manuscript to PLOS ONE. The manuscript has greatly improved and virtually all the issues have been resolved with one exception of "overlooked" conclusion adjustment as noted by the Reviewer 2. Therefore, we invite you to complete the final round of revisions and submit the revised version of the manuscript that addresses the points raised during the review process.

We would appreciate receiving your revised manuscript by Nov 22 2019 11:59PM. To enhance the reproducibility of your results, we recommend that if applicable you deposit your laboratory protocols in protocols.io, where a protocol can be assigned its own identifier (DOI) such that it can be cited independently in the future. For instructions see: http://journals.plos.org/plosone/s/submission-guidelines#loc-laboratory-protocols

We look forward to receiving your revised manuscript.

Kind regards,

Michal A Olszewski, DVM, PhD

Academic Editor

PLOS ONE

Reviewers' comments:

Reviewer's Responses to Questions

**Comments to the Author**

1. If the authors have adequately addressed your comments raised in a previous round of review and you feel that this manuscript is now acceptable for publication, you may indicate that here to bypass the “Comments to the Author” section, enter your conflict of interest statement in the “Confidential to Editor” section, and submit your "Accept" recommendation.

Reviewer #1: All comments have been addressed

Reviewer #2: (No Response)

2. Is the manuscript technically sound, and do the data support the conclusions?

Reviewer #1: Yes

Reviewer #2: Yes

3. Has the statistical analysis been performed appropriately and rigorously? 

Reviewer #1: Yes

Reviewer #2: Yes

4. Have the authors made all data underlying the findings in their manuscript fully available?

Reviewer #1: Yes

Reviewer #2: Yes

5. Is the manuscript presented in an intelligible fashion and written in standard English?

Reviewer #1: Yes

Reviewer #2: (No Response)

6. Review Comments to the Author

Reviewer #1: (No Response)

Reviewer #2: My comment 2 is not addressed in the revised manuscript. Please change the conlusions on page 3.

I would suggest to modify the conclusions to: “Macmoondongtang treatment alleviates asthma symptoms and modulate the Th1-/Th2- related cytokines. Glycyrrhizin and liquiritin could be the major the active therapeutic components”.

7. PLOS authors have the option to publish the peer review history of their article (what does this mean?). If published, this will include your full peer review and any attached files.

Reviewer #1: No

Reviewer #2: No

---

## [Author Response · Author response to Decision Letter 2]

10 Oct 2019

Reviewer #2: 

1. My comment 2 is not addressed in the revised manuscript. Please change the conlusions on page 3. I would suggest to modify the conclusions to: “Macmoondongtang treatment alleviates asthma symptoms and modulate the Th1-/Th2- related cytokines. Glycyrrhizin and liquiritin could be the major the active therapeutic components”.

Ans) Thank you so much for your suggestion and I amended the conclusion part in the abstract.

---

## [Editor Report · Decision Letter 3]

16 Oct 2019

Macmoondongtang modulates Th1-/Th2-related cytokines and alleviates asthma in a murine model

PONE-D-19-17023R3

Dear Dr. Park,

We are pleased to inform you that your manuscript has been judged scientifically suitable for publication and will be formally accepted for publication once it complies with all outstanding technical requirements.

With kind regards,

Michal A Olszewski, DVM, PhD

Academic Editor

PLOS ONE
---

## [Editor Report · Acceptance letter]

14 Nov 2019

PONE-D-19-17023R3 

Macmoondongtang modulates Th1-/Th2-related cytokines and alleviates asthma in a murine model 

Dear Dr. Park:

I am pleased to inform you that your manuscript has been deemed suitable for publication in PLOS ONE. Congratulations! Your manuscript is now with our production department. 

With kind regards,

on behalf of

Dr. Michal A Olszewski 

Academic Editor

PLOS ONE